# Identification of Genomic Regions Associated with Vine Growth and Plant Height of Soybean

**DOI:** 10.3390/ijms23105823

**Published:** 2022-05-22

**Authors:** Yipeng Lu, Jiaming Zhang, Xiaoyang Guo, Jingjing Chen, Ruzhen Chang, Rongxia Guan, Lijuan Qiu

**Affiliations:** National Key Facility for Crop Gene Resources and Genetic Improvement, Institute of Crop Sciences, Chinese Academy of Agricultural Sciences, Beijing 100081, China; luyipeng@genetics.ac.cn (Y.L.); zjm15545596956@163.com (J.Z.); guoxiaoyanggxy@163.com (X.G.); jingjing91627@126.com (J.C.); rx_guan@sina.com (R.C.)

**Keywords:** wild soybean, vining growth, plant height, QTL

## Abstract

Vining growth (VG) and high plant height (PH) are the physiological traits of wild soybean that preclude their utilization for domesticated soybean breeding and improvement. To identify VG- and PH-related quantitative trait loci (QTLs) in different genetic resources, two populations of recombinant inbred lines (RILs) were developed by crossing a cultivated soybean, Zhonghuang39 (ZH39), with two wild soybean accessions, NY27-38 and NY36-87. Each line from the two crosses was evaluated for VG and PH. Three QTLs for VG and three for PH, detected in the ZH39 × NY27-38 population of the RILs, co-located on chromosomes 2, 17 and 19. The VG- and PH-related QTL in the ZH39 × NY36-87 population co-located on chromosome 19. A common QTL shared by the two populations was located on chromosome 19, suggesting that this major QTL was consistently selected for in different genetic backgrounds. The results suggest that different loci are involved in the domestication or adaptations of soybean of various genetic backgrounds. The molecular markers presented here would benefit the fine mapping and cloning of candidate genes underlying the VG and PH co-localized regions and thus facilitate the utilization of wild resources in breeding by avoiding undesirable traits.

## 1. Introduction

Cultivated soybean [*Glycine max* (L.) Merr.] was domesticated from its progenitor wild soybean (*Glycine soja* Sieb. & Zucc.). Wild soybean is a valuable genetic resource for soybean improvement because of its many agronomically desired traits, including high protein and biotic and abiotic stress tolerances [1,2,3,4,5,6]. However, due the presence of undesired traits, such as the indeterminate vine-like habit, lodging and small seed size, a simple cross of *G. max* and *G. soja* is unlikely to result in progeny with only the desirable agronomic traits. Often, we need two to three backcrosses with the *G. max* parent to obtain agronomically acceptable segregates [7,8]. Thus, research on the genetic controls of domestication-related traits and unwanted traits would help further the utilization of wild genetic resources in soybean breeding.

Researchers have attempted to evaluate the complex genetic inheritance of undesired traits in domesticating soybean, including the trait of a vine-like growth habit. Two QTLs of twining habit, *qTH-D1b* and *qTH-G*, have been mapped on chromosomes 2 and 18 of a population of 96 RILs from a cross of Tokei 780 (*G. max*) × Hidaka 4 (*G. soja*) [9]. A total of 132 domestication-related QTLs were detected in two populations derived from Williams 82 × PI 468916 (*G. soja*) and Williams 82 × PI 479752 (*G. soja*) through genotyping sequence data, and among the 12 QTLs related to growth habit, only *qGH-19-2* (with PVE values of 5% and 10%) was detected in both populations [10]. In an experiment that evaluated the fitness of hybrids between *G. max* cultivars and *G. soja*, a common QTL in the linkage group L related to stem length was detected in two segregating populations across two distinct environments representing southern and northern fields in Japan [11]. The QTLs related to VG varied among the different backgrounds and development stages. Seven QTLs were mapped on chromosomes 1, 13, 18 and 19 in the flowering stage (R1), and two QTLs were mapped on chromosomes 18 and 19 in the maturity stage, of which *qVGH-18-2* was mapped in two RIL populations derived from a common wild soybean PI342618B and two cultivated soybeans [12]. In a recombinant inbred population derived from the cultivated soybean Union and the wild soybean W05, growth-habit-related QTLs were mapped on chromosomes 13 and 18, and shoot-length-related QTLs were mapped on the other six chromosomes [13]. To date, only two candidate genes of VG have been reported [12,13]. A gibberellin oxidase (GAox) gene, *Glyma18g06870*, was proposed as the candidate of *VGH1* within the *qVGH-18-2* locus, and this gene was assumed to play a role in soybean growth habit based on the large *F*_ST_ value of *Glyma18g06870* observed between the erect and viny soybean populations [12]. The variation in the copy number of the second gene, *gibberellin 2-oxidase 8A/B* (*GA2ox8*), was proposed to correspond to growth habit via enabling gene expression levels, and the genomic region on chromosome 13 likely underwent strong selection during the soybean domestication [13].

Bulked segregant analysis (BSA) was developed for the rapid identification of markers linked with qualitative loci using molecular markers and DNA bulked from individuals with contrasting phenotypes [14]. With the development of next generation resequencing technology, strategies that combine BSA with high-throughput genome sequencing (BSA-seq and MutMap) have been gradually used to quickly identify QTLs related to various traits in crops [15,16,17,18]. Eight consistent QTLs related to soybean plant height and the number of nodes have been identified in populations with chromosome segment substitutions by using BSA-seq [19]. Recently, a strategy combining BSA with a high density single nucleotide polymorphism (SNP) array has been successfully used for mapping yield-, disease resistance- and salt-tolerance-related QTLs in crops [19,20,21,22,23,24]. Research into the genetic causes of soybean vining growth and plant height has shown polygenic architecture that involves a number of loci. In order to quickly access the genetic architecture of VG and PH in different genetic backgrounds, here, we conducted BSA with an SNP chip analysis in two soybean RIL populations generated from crosses of the soybean cultivar Zhonghuang39 (ZH39) and the two wild soybean NY27-38 and NY36-87. Simple sequence repeat (SSR) markers were used to genotype the RIL populations to confirm the genomic regions of the QTLs.

## 2. Results

### 2.1. Plant Height and Vine-like Growth of Parents and RILs

The RILs derived from ZH39 × NY27-38 (RIZN38) and ZH39 × NY36-87 (RIZN87) and the parental plants were planted in Sanya. The ZH39 and *G. soja* parents showed significant differences in terms of PH and VG (*p* < 0.05). The mean height of the erect stems of ZH39 was 27.7 cm. The vining wild soybeans NY27-38 and NY36-87 had mean winding numbers of 12.5 and 25.6, respectively, and mean PH values of 31.7 and 81.9 cm, respectively (Figure 1A–C). 

In the RIZN87 population, 27.6% of lines (60 RILs) were erect, and the other 157 lines were vining and had a winding number less than that of the NY36-87 population (Figure 1D). In the RIZN38 population in 2017, 53.4% of lines (117 RILs) were erect, while only 1.8% of lines (4 RILs) were wound more times around the plant-supporting strings than those of NY27-38, with the highest number at 16.9 in 2017 (Figure 1F). The percentage of erect RILs (108 lines) in the RIZN38 population in 2018 was similar to that in 2017, while there was a slightly higher number of lines (15 RILs) that had a greater winding number than that of NY27-38 (Figure 1H) in 2018 than in 2017. The PH of 90.3% of the RIZN87 lines fell in between the heights of the parents, and only 4.6% and 5.1% of the lines were shorter than ZH39 and taller than NY36-87, respectively (Figure 1E). Compared with the PH in RIZN87, the PH in the RIZN38 population showed transgressive segregation, with few RILs (21 RILs in 2017 and 18 RILs in 2018) having a PH between those of the parents (Figure 1G,I). 

PH was significantly correlated with VG in the RIZN87 population (*r =* 0.96, *p* < 0.001). In the RIZN38 population, PH was significantly correlated with VG in both years, as indicated by the coefficients of 0.95 in 2017 and 0.89 in 2018 (*p* < 0.001). In the RIZN38 population, the correlation coefficient of PH in 2017 and 2018 was 0.71, and that of VG was 0.72 in 2017 and 2018.

### 2.2. QTL Analysis of VG and PH in the ZH39 × NY27-38 Population

A total of 114472 homozygous polymorphic SNPs were found between ZH39 and NY27-38, of which 545 showed polymorphism between the erect EB2 and the vining VB2 bulks. There were 140 and 142 polymorphic SNPs in chromosomes 17 and 19, respectively, and 4–34 SNPs distributed in each of the other 18 chromosomes (Figure 2A), indicating that the VG- and PH-related QTLs were likely located on chromosomes 17 and 19. The maximum number of different SNPs, 79, was found in the 42–46 Mb region of Chr 19, while moderate SNP numbers of 14, 3, 14 and 9 were found in regions 42–46 Mb of Chr 2, 38–42 Mb of Chr 9, 36–40 Mb of Chr 13 and 2–6 Mb of Chr 15, respectively (Figure 2B). 

To validate the QTLs of VG and PH in the RIZN38 population, a genetic linkage map spanning 2388.6 cM was constructed by genotyping 219 RILs using 200 polymorphic SSR markers. QTL mapping was performed using the phenotype data on VG and PH collected in 2017 and 2018. Three QTLs related to VG, *qVG-2*, *qVG-17* and *qVG-19*, were mapped on chromosomes 2, 17 and 19, respectively, for both years of study. The PH-related QTLs, *qPH-2*, *qPH-17* and *qPH-19*, correspondingly mapped to the same locations as the QTLs for VG. The QTL on Chr 19 was a major one related to its respective traits, VG and PH, which accounted for 15.30% and 17.43% of the phenotype variation for VG and PH, respectively, in 2017 and 15.66% and 22.01% of the variation for VG and PH, respectively, in 2018. The other QTLs had small effects on VG and PH. The PVE (phenotypic variation explained) values of the QTL on Chr 2 were 5.80% for VG and 3.96% for PH in 2017, and they were 9.84% for VG and 6.33% for PH in 2018. The PVEs of the QTL on Chr 17 were 9.81% for VG and 14.61% for PH in 2017, and they were 4.01% for VG and 8.52% for PH in 2018. ZH39 contributed to reducing VG and PH at the loci on chromosomes 2 and 19, while ZH39 contributed to increasing VG and PH at the locus on Chr 17 (Figure 2C, Table 1). The linkage map analysis verified the two major QTLs identified using BSA, and it also identified a minor QTL on Chr 2 for both VG and PH (Table 1).

### 2.3. QTL Analysis for VG and PH in Population of ZH39 × NY36-87

Two DNA bulks, EB1 and EB1, from the RIZN87 population were genotyped by a ZDX1 SNP array. Of the 89106 SNPs that showed polymorphism between the two parents, 1027 SNPs were polymorphic between the two bulks. Most of the polymorphic SNPs (73.2%) were distributed on chromosomes 7 (195), 12 (157), 13 (130) and 19 (270), the majority of which were distributed within the regions 18–36 Mb, 18–34 Mb, 24–30 Mb and 4–44 Mb of the four chromosomes, respectively (Figure 3A,B). These results indicated that potential PH- and VG-related QTLs were located in these four chromosomes.

To confirm the existence of VG- and PH-related QTLs in these chromosomes, we screened the 50 individuals with extreme phenotypes used to construct the bulks EB1 and VB1 by an SNP array. The 89,106 polymorphic markers were converted into 1352 bin markers. The QTLs on Chr 19 were identified for VG and PH. The percentages of PVE of *qVG-19.1* and *qPH-19.1* were 56.04% and 52.26%, respectively (Figure 3C, Table 2). Two polymorphic SSR markers, SSR_19_1323 and SSR_19_1335, on Chr 19 were selected to confirm the QTL in the 217 RILs of the RIZN87 population. The PVEs of *qVG-19.1* and *qPH-19.1* were 6.14% and 6.07%, respectively (Table 2). No VG- and PH-related QTLs were identified on chromosomes 7, 12 and 13 in the screening of 50 individuals with extreme phenotypes. This was also confirmed by using the SSR markers flanking the intervals where the polymorphic SNPs were distributed (Chr 7, 18–36 Mb; Chr12, 18–34 Mb; Chr 13, 24–30 Mb) to genotype the 217 RILs of RIZN87. These results showed that the major QTL can be identified by genotyping individuals with extreme phenotypes (Figure 2, Table 2).

### 2.4. Phenotypic Effect of QTLs Associated with VG

To determine the phenotypic effect of the VG-related QTLs identified in the RIZN38 population, the phenotypes of lines were analyzed based on the three-loci genotypic combination at three SSR loci that closely linked with *qVG-2*, *qVG-17* and *qVG-19*. Eight genotypic combinations were found in the population at the three VG-related QTLs. Where *qVG-19* was fixed for the ZH39 allele (AA), the RILs were all erect regardless of their genotype at the loci of *qVG-2* and *qVG-17* (Figure 4A), suggesting that the major QTL *qVG-19* may have epistatic interactions with *qVG-2* and *qVG-17* in VG. In the NY27-38 genetic background at *qVG-19* (*qVG-19*^BB^), the combinations of *qVG-2*^AA^ and *qVG-17*^BB^, *qVG-2*^AA^ and *qVG-17*^AA^ and *qVG-2*^BB^ and *qVG-17*^BB^ differed significantly in the measure of VG, with the highest value obtained where the *qVG-19*^BB^, *qVG-2*^BB^ and *qVG-17*^AA^ alleles were present at the three loci (Figure 4A). These results suggest that there are interaction effects among the three QTLs (Figure 4A). In the RIZN87 population, the lines with the BB genotype showed greater VG than the lines with the AA genotype at the locus *qVG-19.1* (Figure 4B).

### 2.5. Genetic and Phenotypic Variation of Near Isogenic Lines (NILs)

Two NILs, 18NG487 and 18NG488, were genotyped using an SNP array. We identified 283 homozygous polymorphic SNPs, indicating that the background similarity between 18NG487 and 18NG488 was 99.81%. The 283 polymorphic SNPs were distributed on chromosomes 5, 17 and 19, with 91.2% (258 of 283) concentrated in the 40–50 Mb interval of Chr 19, where *qVG-19* was located (Figure 5A,B). Fourteen SNPs were located in the 35–40 Mb interval of Chr 5, and 11 SNPs were in the 35–40 Mb interval of Chr 17 (Figure 5B). The phenotypes of the NILs were examined in 2018, and both 18NG487 and 18NG488 produced main stems; the stems of 18NG487 were erect and the stems of 18NG488 showed VG (Figure 5C). The VG and PH values of 18NG487 were 0 and 17 cm, respectively, while the VG and PH values of 18NG488 were 5.5 and 40.92 cm, respectively (Figure 5D,E).

## 3. Discussion

*Glycine max* and *G. soja* are morphologically distinct species; *G. max* displays an upright-bush type of stem growth, while *G. soja* shows indeterminate vine-like growth [25,26]. To evaluate the QTLs of the soybean architecture, we made two RIL populations derived from *G. max* × *G. soja*, performed QTL mapping in the two populations and identified the major conserved QTLs in the different genetic backgrounds.

By using a combination of BSA and SNP genotyping arrays, stripe rust resistance loci have been successfully mapped on wheat chromosomes 2B and 1B, respectively [24,27]. In rice, 21 and 34 QTL regions were identified in salt tolerant and sensitive bulks, respectively, genotyped with a 50 K SNP chip [23]. Here, by using BSA with a high-density SNP array, the genomic regions associated with VG and PH were identified on chromosomes 17 (12–28 Mb) and 19 (41–44 Mb) in the RIZN38 population. These two regions were confirmed in the RILs by genetic mapping, and two additional QTLs of VG and PH (with smaller PVE values) on chromosome 2 were identified. We identified 23 homologous SNPs between the DNA bulks VB2 and EB2, of which 69.6% (16 of 23) were distributed within the 41–44 Mb interval in which the QTLs *qVG-2* and *qPH-2* were mapped (Figure 2C, Table 1). This confirmed that the method of combining BSA with an SNP array can precisely identify major QTLs.

In the RIZN87 population, the homozygous SNPs between the parents and the DNA bulks VB1 and EB1 were mainly located on chromosomes 7, 12, 13 and 19 (Figure 3A,B). However, only one of the potential regions on Chr 19 containing a QTL of interest was confirmed by genotyping 50 extreme RILs (Figure 3C, Table 2). The identification of false positive loci may due to (1) the smaller pooled sample size of 22 RILs or (2) the limitations of the homozygote allele-calling algorithm of the SNP array, especially if the candidate region is associated with a minor-effect QTL. Magwene et al. [28] suggest that 15–20% of the population of interest be used when bulking in BSA-seq. According to an experiment conducted on rice, 30 RILs should be used when bulking to achieve greater than 90% accuracy in determining the amount of heterogeneity [23]. The QTL peaks at chromosomes 19 and 13 were centered at the 4 Mb and 2 Mb intervals in the RNZN38 population, while the SNPs lacked obvious peaks on chromosomes 2 and 12 (Figure 3B). Furthermore, only a QTL region on Chr 19 was identified by screening 50 extreme RILs using an SNP array, indicating that screening extreme phenotypes is a more effective way to expedite QTL discovery than using BSA combined with an SNP array. 

The genetic controls of traits as a result of domestication are more simple in crops such as maize, common bean and barley than in soybean because their domesticated traits are only controlled by a small number of conserved QTL [29,30,31,32,33]. One of the conserved growth-habit-related QTLs was mapped in the 4.9–5.9 Mb interval of Chr 18 in soybean [10,13]. In the 227 kb region of *qVGH18-2*, *Glyma18g06870* revealed considerable differentiation between 18 vine and 14 erect soybeans, and thus, it was proposed as the corresponding gene of *qVGH18-2* and named *VGH1* [12]. Another conserved VG-related locus was mapped within the 37.8–39.2 Mb region on Chr 13 of the Wm 82 genome and within the 46.1–46.5 Mb region on Chr 11 of the W05 genome [13]. A closer examination of the genomic variation revealed variation in the copy numbers of the *GA2ox8* genes between Wm82 and W05. The overexpression of the candidate genes of *GA2ox8* significantly limited growth in PH, while the CRISPR/Cas9-induced mutants showed longer internode lengths. Together, these results indicate that the copy number variation in *GA2ox8* underlies the regulation of the internode and shoot lengths of soybean [13].

The VG- and PH-related QTLs detected in our NIZN38 population, *qVG-2* and *qPH-2*, have also been reported in previous studies, indicating a conserved locus in different genetic backgrounds. A related QTL, *qTH-D1b*, was mapped near Satt546 and accounted for 20.5% of the PVE of a population of 96 RILs examined in a one-year experiment [9]. In a population derived from the cross Williams 82 × PI479752, one of the ten QTLs associated with growth habit was mapped on Chr 2 at 44.6 Mb, with a PVE value of 10.1% [10]. However, no QTL related to PH or stem length was detected in the same interval by the researchers of these two reports.

The region of the VG-related QTLs *qVG-19* and *qVG-19.1* in both NIZN38 and NIZN87 was located within 44.7–45.3 Mb on Chr 19. One of the growth-habit-related loci, *qGH-19-2*, was mapped in the same position on Chr 19 of each of the two populations, with a PVE value of 5.1% for WP479 and of 9.8% for WP469 [10]. This locus was also related to stem length and accounted for 19.4% and 36.9% of the PVE in the two populations (JP036034 × Ryuhou and JP110755 × Fukuyutaka, respectively), and the candidate gene was deduced as the soybean indeterminate growth habit gene *Dt1* [11]. Five QTLs related to the vine growth habit were detected on chromosomes 18 and 19 at the maturity stage (R8) of soybean. Among these five, *qVGH-19-3* was mapped to the same position on Chr 19 as *qVG19* [12]. However, at *qGH-19-2* and *qVGH-19-3*, the candidate gene caused a reduction in the vining growth of plants, so the gene is unlikely to be *Dt1* because the *G. max* parents used in these populations are indeterminate with the *Dt1* allele, as are the *G. soja* parents [10,34]. In the research of Liu et al. [9], no significant correlation was observed between the data representing the determinate habit and those representing the twinning habit, and only a QTL related to maximum internode length was observed at the *Dt1* locus. Fine mapping is still required to determine if *Dt1* is allelic to *qPH19* and *qPH19-1* in our future research, because the *Dt1* allele can result in 45–60% lower soybean PH [35]. Other genes linked to *Dt1* could also be candidate genes of the QTLs of VG, as was observed in the maize *tb1* locus, where a QTL related to ear morphology was proved to be multiple linked QTLs [36]. Similarly, the work of Lemmon and Doebley [37] characterized a domestication-trait-related region on maize chromosome 5 and revealed several linked QTLs of small effect. In common bean, the twining-related gene *Tor* was reported to be correlated with the determinacy gene *fin* (*PvTFL1y*), indicating either a pleiotropic effect of *PvTFL1y* or the action of closely linked genes corresponding to the phenotype [31,38].

It is curious that an allele in ZH39 at the locus *qVG-17* may have promoted soybean VG despite the fact that ZH39 is not known to exhibit the vining growth phenotype. One stem-length-related QTL was also mapped to this region in the BC_2_F_2_ populations derived from the JP036034 × Ryuhou cross and in the F_2_ population derived from the JP110755 × Fukuyutaka cross. Notably, similar to our surprising result, the parental cultivars Ryuhou and Fukuyutaka, rather than the wild parent, provided the alleles that positively affected long stem growth [11]. These observations suggest that cultivated soybean, such as ZH39, Ryuhou and Fukuyutaka, may have loci regulating stem growth that are different from those loci identified in other experiments. This is supported by the fact that the alleles from Williams 82 at the loci *qGH-12* and *qGH-15* exhibited positive effects on VG in a Williams 82 × PI479752 population [10]. It is unknown whether these loci simply differentiated from the parents used in the specific experiment or if they were involved, at least partially, in soybean domestication.

Undoubtedly, the vining growth and plant height in soybean are regulated by many genes with a small effect which are typically polygenic, just like those related to complex abiotic stress tolerance traits [39]. However, major vining growth related QTLs with an effect of 10% or more were identified in different soybean backgrounds, which made it possible to select against vining growth through marker-assisted selection (MAS) coupled with backcrossing in future soybean breeding programs for transferring the elite traits from wild relatives [40].

To better elucidate the genetic relationships throughout soybean domestication, efforts are needed to identify more loci related with plant architecture to enhance the utilization of wild soybean resources. Genome-wide association study (GWAS) should be used for VG analysis based on the evaluation of the phenotypes and genotypes of representative soybean panels, as was conducted for soybean cyst nematode resistance [41]. To clone the genes underlying these loci, a chromosome segment substitution lines (CSSL) population should be constructed for the fine mapping of each related QTL [19]. Precise genome editing techniques will benefit the utilization of genes that control plant architecture in a selection of ideal varieties [42], provide strategies to reduce gene pleiotropy through the modification of *cis*-regulatory regions [43] or manipulate target genes for the de novo domestication of wild crop relatives [44].

In the present study, we combined BSA analysis, linkage mapping and the genotyping of individuals with extreme phenotypes by an SNP array for the QTL mapping of two populations. Three QTLs associated with VG and PH were detected and co-located on chromosomes 2, 17 and 19, among which *qVG-19* and *qPH-19* were identified as major QTLs in our two RIL populations. In comparison with BSA, the screening of phenotypically extreme individuals was an effective way to identify QTLs while obtaining fewer false positive loci. The fine mapping of these QTLs is needed for the cloning of the underlying genes, which will broaden our understanding of the genetic mechanisms of vining growth in soybean.

## 4. Materials and Methods

### 4.1. Plant Materials and Phenotype Investigation

The soybean cultivar ZH39 and the wild soybean NY27-38 and NY36-87 were obtained from the Chinese Academy of Agricultural Sciences. ZH39 has an erect growth habit that does not twine and is widely grown in northern China, while NY27-38 and NY36-87 are two typical vining wild accessions collected from the Hebei and Henan provinces of China. Two crosses were made between ZH39 (the female parent) and the wild soybean NY27-38 and NY36-87 in 2013. The recombinant inbred line (RIL) populations were advanced from F_2_ to F_6_ and F_7_ through single-seed descent. The two populations, ZH39 × NY36-87 (RIZN87) and ZH39 × NY27-38 (RIZN38), produced 217 F_7_ lines and 219 F_6_ lines, respectively.

To evaluate the phenotypes, all of the plants were grown in an experiment station at Sanya, Hainan Island (18.4°N, 109.2°E). The 217 F_7_ lines derived from RIZN87 were planted in 2017, while the 219 F_6_ RILs derived from RIZN38 were planted in 2017 and 2018. All the seeds were scarified before sowing to promote germination. Each RIL was planted in a single row plot that was 150 cm long and had a 35 cm row space, and 15 seeds were planted for each RIL. The plant height and vining trait data were collected at the maturity stage from 10 individual plants for each line. The mean values were calculated as the final trait value for each line. The plant height was the length of the main stem (measured from the soil surface to the apical tip). The vining trait was represented by the number of times the main stem wound itself around a thin string used to support the plants (henceforth, the mean counts are referred to as the winding number).

### 4.2. Genotyping DNA Bulks and Individuals with Extreme Phenotypes by SNP Array

The genomic DNA were extracted from the young soybean leaves using a DNA purification kit K0512 (Thermo Fisher ScientificBaltics, Vilnius, Lithuania). The vining bulk (VB1) and erect bulk (EB1) were generated by pooling 200 ng of DNA from 25 vining and 25 erect lines, respectively, from the RIZN87 population. We considered only the ‘extreme’ phenotypes (individuals at the two tails of the phenotypic distribution of traits) for bulking and thus selected individuals with a winding number of 0 and a PH of less than 10 cm for the bulking of erect lines and individuals with a winding number greater than 10 and a PH higher than 65 cm for the bulking of vining lines. Of the RIZN38 population, the vining (VB2) and erect (EB2) bulks consisted of 200 ng of DNA pooled from 22 vining and 22 erect lines, respectively. The selection criteria for the erect RIZN38 plants were winding number = 0 and PH < 10 cm, while the selection criteria for the vining RIZN38 plants were winding number < 10 and PH < 40 cm. These bulks and the 50 individuals used to establish the VB1 and EB1 bulks, as well as the three parental lines, were genotyped using the ‘Zhongdouxin No.1′ (ZDX1) BeadChip, with 159,072 SNPs processed through the Illumina iScan platform (Illumina, Inc., San Diego, CA, USA) [45]. The SNP alleles were called using the GenomeStudio v2.0.5 Genotyping software (V2011.1, Illumina, Inc., San Diego, CA, USA).

The SNP information of those bulks was obtained and filtered, and the filtering process was as follows: (1) delete the heterozygous sites between the parents, (2) filter out the non-polymorphic sites between the parents, (3) filter out the non-polymorphic sites between the erect DNA bulk (EB) and vining DNA bulk (VB) and (4) filter out the heterozygous sites between the VB and EB.

### 4.3. Genotyping RIZN38 Individuals with SSR Markers

A total of 864 SSR markers distributed on 20 chromosomes and selected from SoyBase (https://soybase.org, accessed on 15 April 2017) were used to identify the polymorphic markers between the parents ZH39 and NY27-38. The amplified products were screened in 6% denaturing polyacrylamide gels. We recorded the alleles at each locus. The allele from the female parent was labeled ‘A’, and that from the male parent was ‘B’. The heterozygous alleles were ‘H’, and the missing data were ‘X’.

Two hundred polymorphic SSR markers were used to amplify the individuals of the RIZN38 population. To confirm the candidate QTLs identified by the BSA-SNP array in RIZN87, the polymorphic SSR markers located in these regions were used to screen the RIL populations. IciMapping4.1 was used to construct a genetic linkage map with inclusive composite interval mapping [46]. The threshold of the logarithm of the odds (LOD) was higher than 2.5, and the scanning step distance was 1 cM [47].

### 4.4. Development of Near Isogenic Lines (NILs)

To develop NILs that differed only for the target traits, we selected a plant from an F_5_ population (ZH39 × NY27-38) that was heterozygous within the target region by using SSR markers flanking the major VG-related QTL on chromosome 19. A selected F_5_ plant was self-pollinated for three generations (from the F_6_ generation to the F_8_ generation) to obtain a pair of erect and vining NILs: 18NG487 and 18NG488, respectively. The DNA of 18NG487 and 18NG488 were screened by the ZDX1 SNP array to evaluate for genetic background similarity.

## Figures and Tables

**Figure 1 ijms-23-05823-f001:**
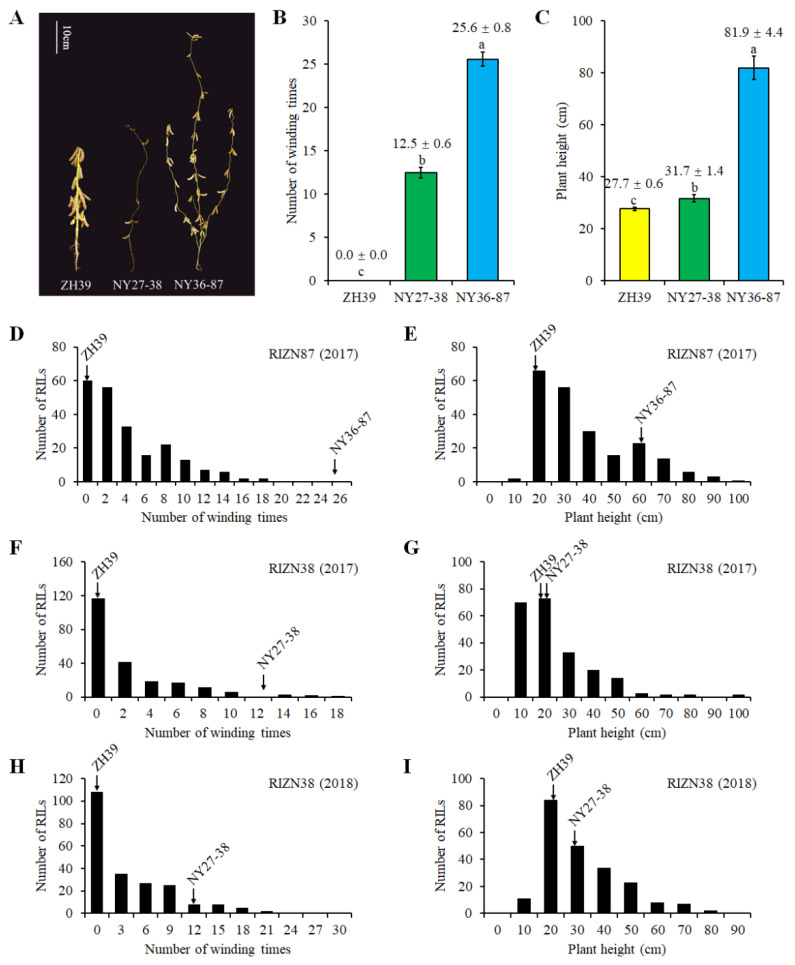
Phenotype and frequency distributions of the vining growth (VG) and plant height (PH) of RIL populations from crosses of ZH39 × NY36-87 (*n* = 217) and ZH39 × NY27-38 (*n* = 219). (**A**) ZH39, NY27-38 and NY36-87 phenotypes. The number of times plant stems wound themselves around a supporting structure (**B**) and the plant height (**C**) of ZH39, NY27-38 and NY36-87, *n* = 15. Different letters above the bars indicate significant differences between parents (one-way ANOVA followed by Fisher’s test, *p* < 0.05). The 2017 (**D**) VG and (**E**) PH data of an RIL population derived from ZH39 × NY36-87, the (**F**) 2017 and (**G**) 2018 VG data of a ZH39 × NY27-38 RIL population and the (**H**) 2017 and (**I**) 2018 plant height data of a ZH39 × NY27-38 RIL population.

**Figure 2 ijms-23-05823-f002:**
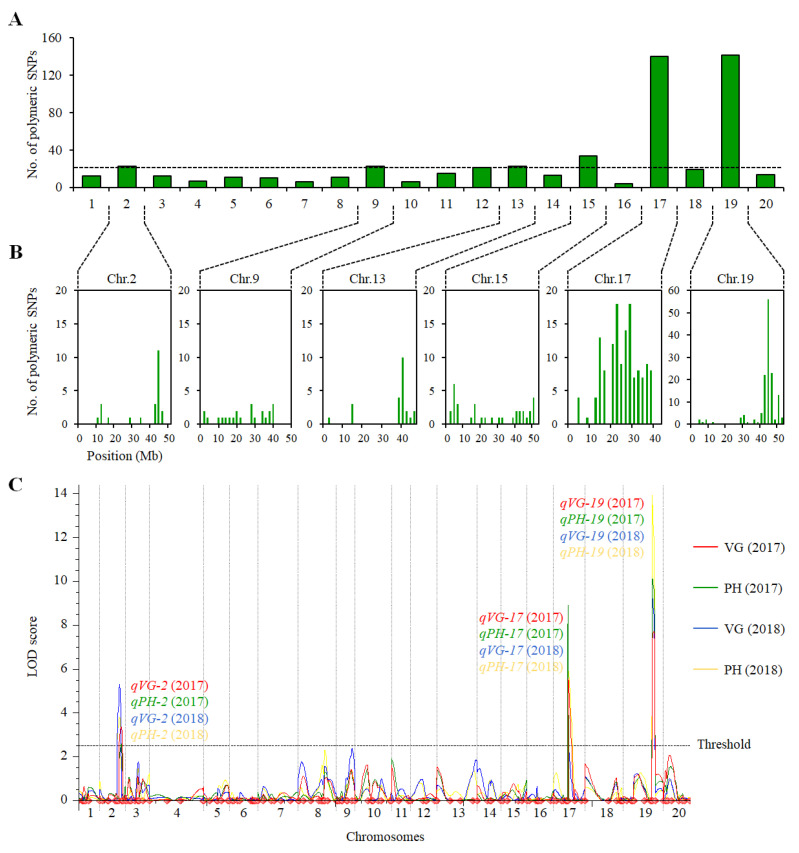
Number of polymorphic SNPs and QTL mapping in the RIL population from the ZH39 × NY27-38 cross. (**A**) The number of polymorphic SNPs between two DNA bulks, EB2 and VB2, on all chromosomes. The dotted line is the threshold to identify putative QTLs. (**B**) Six putative regions for QTLs associated with VG and PH. Each bar represents the number of polymorphic SNPs between EB2 and VB2 in a 2 Mb region. (**C**) QTL mapping of VG (red for 2017 and blue for 2018) and PH (green for 2017 and yellow for 2018) in the ZH39×NY27-38 RIL population; the threshold (dotted line) of LOD values was 2.5.

**Figure 3 ijms-23-05823-f003:**
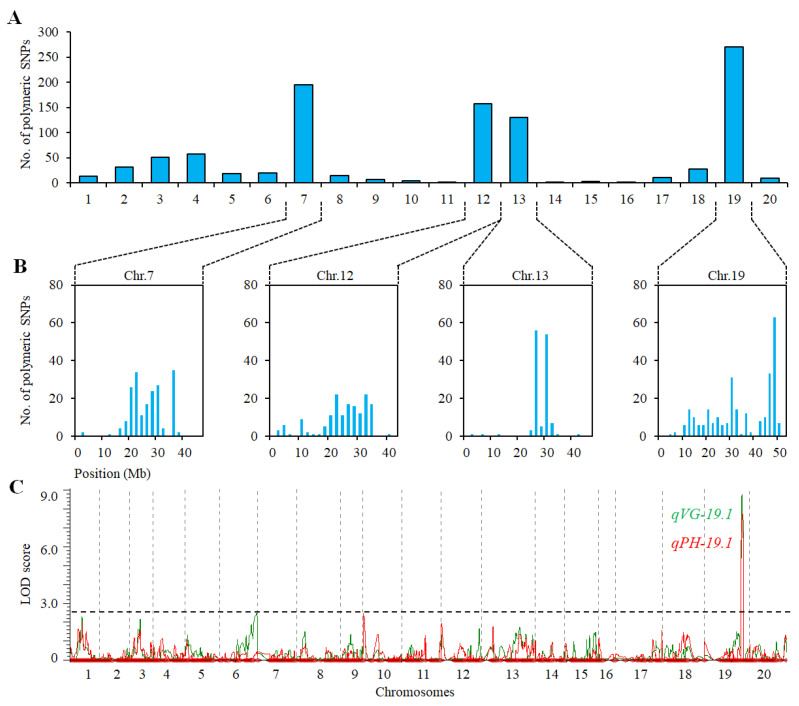
QTL mapping of VG and PH in the ZH39×NY36-87 RIL population. (**A**) Numbers of polymorphic SNPs on all chromosomes between two DNA bulks, EB1 and VB1. (**B**) Four putative regions for QTLs associated with VG and PH; the bars represent the number of polymorphic SNPs between EB2 and VB2 in a 2 Mb region. (**C**) Composite interval QTL mapping for VG and PH by genotyping 50 phenotypically extreme individuals from ZH39 × NY36-87; VG (green) and PH (red).

**Figure 4 ijms-23-05823-f004:**
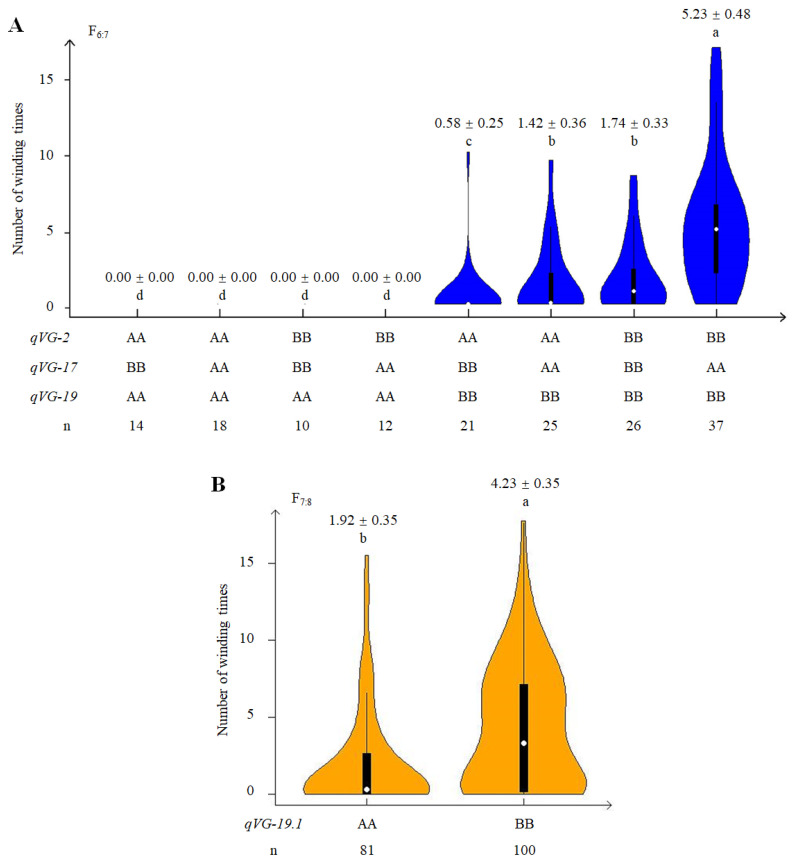
Vining growth of soybean lines with different alleles at related QTLs. (**A**) Violin plots show the VG of RIL families carrying various allele combinations at three QTLs (*qVG-2*, *qVG-17* and *qVG-19*) in the ZH39 × NY27-38 population. (**B**) Violin plots of the VG of RIL families (derived from the ZH39 × NY36-87 cross) carrying parental genic alleles at *qVG-19.1*. AA and BB represent alleles from the *Glycine max* and *G. soja* parents, respectively, at the corresponding SSR loci. Box and whisker plots are shown in each violin plot; the white node in the center indicates the median. Different letters above the plots indicate significant differences between the lines with different genotypes (one-way ANOVA followed by Fisher’s test, *p* < 0.05).

**Figure 5 ijms-23-05823-f005:**
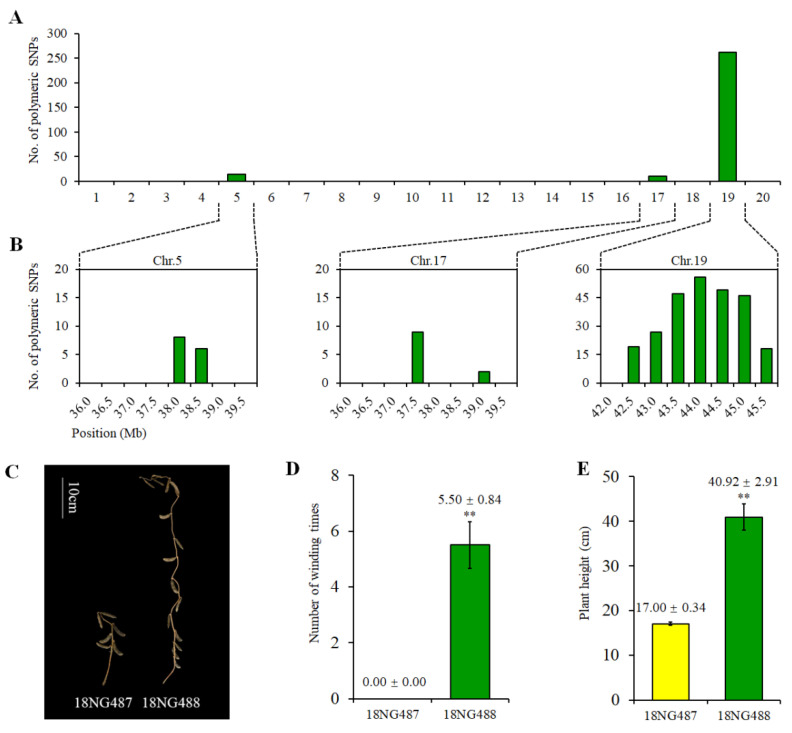
Genotype and phenotype evaluations of the NILs with erectness (18NG487) and vining growth (18NG488). (**A**) The number of polymorphic SNPs on all chromosomes between the two NILs. (**B**) The number of polymorphic SNPs on chromosomes 5, 17 and 19 of the two NILs; each bar represents the number of polymorphic SNPs between the NILs in a 5 Kb region. Phenotypic illustration (**C**), VG (**D**) and PH (**E**) of 18NG487 and 18NG488. Data are mean ± s.d., *n* = 15; ** *p* < 0.01 (Student’s *t*-test).

**Table 1 ijms-23-05823-t001:** QTLs statistics associated with VG and PH in the ZH39 × NY27-38 F_6_ population.

Year	QTLs	Chr.	Marker Interval	Position (Mb)	LOD	PVE (%)	ADD
2017	*qVG-2*	2	SSR_2_1540-SSR_2_1602	43.3–44.3	3.34	5.80	−0.73
	*qVG-17*	17	SSR_2_661-SSR_2_669	11.3–11.6	5.56	9.81	0.96
	*qVG-19*	19	SSR_19_1323-SSR_19_1335	45.1–45.3	7.69	15.30	−1.12
2018	*qVG-2*	2	SSR_2_1540-SSR_2_1602	43.3–44.3	5.31	9.84	−1.54
	*qVG-17*	17	SSR_2_661-SSR_2_669	11.3–11.6	2.63	4.01	1.00
	*qVG-19*	19	SSR_19_1323-SSR_19_1335	45.1–45.3	9.44	15.66	−2.05
2017	*qPH-2*	2	SSR_2_1540-SSR_2_1602	43.2–45.5	2.57	3.96	−2.99
	*qPH-17*	17	SSR_2_661-SSR_2_669	11.3–11.6	8.90	14.61	5.85
	*qPH-19*	19	SSR_19_1323-SSR_19_1335	45.1–45.3	10.13	17.43	−6.54
2018	*qPH-2*	2	SSR_2_1540-SSR_2_1602	43.3–44.3	3.82	6.33	−3.78
	*qPH-17*	17	SSR_2_661-SSR_2_669	11.3–11.6	5.95	8.52	4.45
	*qPH-19*	19	SSR_19_1323-SSR_19_1335	45.1–45.3	13.93	22.01	−7.41

**Table 2 ijms-23-05823-t002:** QTLs statistics associated with VG and PH in the ZH39 × NY36-87 F_7_ population.

Method	QTL	Chr.	Marker Interval	Position (Mb)	LOD	PVE (%)	ADD
SNP array	*qVG-19.1*	19	Gm44946-Gm45005	44.7–45.4	8.79	56.04	−4.38
	*qPH-19.1*	19	Gm44946-Gm45005	44.7–45.4	7.85	52.26	15.23
BSA	*qVG-19.1*	19	SSR_19_1323-SSR_19_1335	45.1–45.3	3.00	6.14	−0.99
	*qPH-19.1*	19	SSR_19_1323-SSR_19_1335	45.1–45.3	2.97	6.07	−4.52

## Data Availability

The data generated and analyzed during the current study are available from the corresponding author on reasonable request.

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
