# Peer review of "Identification of Genomic Regions Associated with Vine Growth and Plant Height of Soybean"

_ijms, 2022, doi:10.3390/ijms23105823_

Round 1

Reviewer 1 Report

The authors present a straightforward study to identify genomic regions associated with soybean vine growth and plant height based on two populations of recombinant inbred lines developed by crossing of cultivated soybean with two wild soybean assessions. Their study indicated that vine growth and plant height associated QTLs are co-located 2, 17, and 19 chromosome. These identified QTLs are important for further study of genes and molecular mechanisms associated with these traits and are helpful in manipulating these traits in the breeding program.

Author Response

Point 1: The authors present a straightforward study to identify genomic regions associated with soybean vine growth and plant height based on two populations of recombinant inbred lines developed by crossing of cultivated soybean with two wild soybean assessions. Their study indicated that vine growth and plant height associated QTLs are co-located 2, 17, and 19 chromosome. These identified QTLs are important for further study of genes and molecular mechanisms associated with these traits and are helpful in manipulating these traits in the breeding program.

Response 1: Thanks for the comments. The fine mapping and map-based cloning of the QTL on chromosome 19 are ongoing which will benefit for the understanding of soybean domestication and de novo demestication.

Reviewer 2 Report

The work by Lu et al provides thoughtful genetic mapping resources for soybean genetic improvement in terms of plant architecture traits (i.e. vine growth and plant height). The main results adequately summarize major RILs-derived QTLs and marker resources. Yet, before recommending acceptance, I request authors to address the following major improvements.

First, although the introduction section properly closes with an explicit goal, I would encourage authors to list explicit research hypotheses (e.g. a polygenic architecture is expected for plant architecture traits in soybean) at the end of the introduction section. This will allow readers focusing on explicit expectations addressed by the manuscript.

Second, please discuss the mapping statistical power in the light of mapping efforts for complex traits (just as morphological traits) in the closely related species common bean. Speicifally, refer to well-known case studies in common bean to map and predict drought (i.e. Front Plant Sci 2018 9:128, and PLoS One 2013 8(5):e62898) and heat (i.e. Front Genet 2019 10:954, and Genes 2021 12:556) tolerance, both of which may correlate with plant architecture and height (e.g. plant architecture may help preventing soil desiccation and elevated night air temperatures). After all, common bean is a diploid model for soybean in terms of mapping resources, as proposed by Mcclean, Lavin, Gepts, and Jackson in 2008 (reference also to be included).

Third, authors should also improve their discussion by concluding whether the genetic bases of vine growth and plant height may be polygenic or rather Mendelian. This will determine the feasibility to boost soybean breeding programs by relying on marker assisted selection (MAS, for “Mendelian” traits) or alternatively genomic selection (GS, for “polygenic” traits, just as discussed at Genes 2021 12:783 in the context of plant breeding for complex traits, a seminal review to be included).

In this regard, authors should expand the discussion on what is the potential contribution for soybean breeding of last-generation approaches such as genomic prediction (i.e. Front Plant Sci 2021 12:624156 in soybean, reviewed more widely here Front Plant Sci 2020 11:583323, citation to be included), genome-wide scans of selection signatures (carefully review by Front Genet 2020 11:564515), machine learning (nicely discussed in Trends Genet 2018 34:301-12, review to be included), and speed breeding (recently covered in Trend Genet 2021 37:1124-36, review also to be included).

The feasibility of these approaches must explicitly be revisited in a new perspectives section, just after the current discussion section. This new section should start by discussing potential caveats of this and previous studies until now, and propose new avenues of research. Authors should envision key experimental, and novel analyses that will assist in the near future further studies on soybean genetic mapping and improvement of plant architecture traits. These sections will facilitate readers to highlight major trends and opportunities.

Author Response

Point 1: The work by Lu et al provides thoughtful genetic mapping resources for soybean genetic improvement in terms of plant architecture traits (i.e. vine growth and plant height). The main results adequately summarize major RILs-derived QTLs and marker resources. Yet, before recommending acceptance, I request authors to address the following major improvements.

Response 1: Thanks for the comments. We have made major improvements of the manuscript point by point.

Point 2: First, although the introduction section properly closes with an explicit goal, I would encourage authors to list explicit research hypotheses (e.g. a polygenic architecture is expected for plant architecture traits in soybean) at the end of the introduction section. This will allow readers focusing on explicit expectations addressed by the manuscript.

Response 2: Thanks for the suggestion. We have added the hypotheses, “Researches into the genetic causes of soybean vining growth and plant height have showed polygenic architecture that involves a number of loci” at the end of the introduction as the reviewer suggested.

Revised portions are shown in line 72-73.

Point 3: Second, please discuss the mapping statistical power in the light of mapping efforts for complex traits (just as morphological traits) in the closely related species common bean. Speicifally, refer to well-known case studies in common bean to map and predict drought (i.e. Front Plant Sci 2018 9:128, and PLoS One 2013 8(5):e62898) and heat (i.e. Front Genet 2019 10:954, and Genes 2021 12:556) tolerance, both of which may correlate with plant architecture and height (e.g. plant architecture may help preventing soil desiccation and elevated night air temperatures). After all, common bean is a diploid model for soybean in terms of mapping resources, as proposed by Mcclean, Lavin, Gepts, and Jackson in 2008 (reference also to be included).

Response 3: Thanks for the comments. The four references (Cortés et al., 2013; Cortés and Blair, 2018; López-Hernández and Cortés, 2019; Builtrago-Bitar et al., 2021) of common bean drought and heat resistance were related to identification of candidate regions or genes by using SNPs through GWAS and genome-environment association (GEA). Thus, the different statistical models that used for population structure analysis were not suitable for the QTL mapping strategy in bi-parental populations. While these are really well-done case studies when GWAS and GEA are used for soybean traits like plant height, vine growth and abiotic stresses in our future research using natural soybean population.

References:

Buitrago-Bitar, M.A.; Cortés, A.J.; López-Hernández, F.; Londoño-Caicedo, J.M.; Muñoz-Florez, J.E.; Muñoz, L.C.; Blair, M.W. Allelic diversity at abiotic stress responsive genes in relationship to ecological drought indices for cultivated tepary bean, Phaseolus acutifolius A. Gray, and its wild relatives. Genes 2021, 12, 556.

Cortés, A.J.; Monserrate, F.A.; Ramírez-Villegas, J.; Madriñán, S.; Blair, M.W. Drought tolerance in wild plant populations: the case of common beans (Phaseolus vulgaris L.). PLoS ONE 2013, 8, e62898.

Cortés, A.J.; Blair, M.W. Genotyping by sequencing and genome-environment associations in wild common bean predict widespread divergent adaptation to drought. Front. Plant Sci. 2018, 9, 128.

López-Hernández, F.; Cortés, A.J. Last-generation genome-environment associations reveal the genetic basis of heat tolerance in common bean (Phaseolus vulgaris L.). Front. Genet. 2019, 10, 954.

Point 4: Third, authors should also improve their discussion by concluding whether the genetic bases of vine growth and plant height may be polygenic or rather Mendelian. This will determine the feasibility to boost soybean breeding programs by relying on marker assisted selection (MAS, for “Mendelian” traits) or alternatively genomic selection (GS, for “polygenic” traits, just as discussed at Genes 2021 12:783 in the context of plant breeding for complex traits, a seminal review to be included).

Response 4: Thanks for the suggestion. We added discussion of the genetic control of vine growth and plant height as the reviewer suggested and the reference has been included. New content was supplemented in the third paragraph from the bottom “It is undoubtedly that vining growth and plant height in soybean are regulated by many genes with low effect, which was typically polygenic as those related to complex abiotic stress tolerance traits (Cortés et al., 2021). However, major vining growth related QTLs with effect of 10% or more were identified in different soybean backgrounds, which made it possible for selected against vining growth through marker-assisted selection (MAS) coupled with backcrossing in future soybean breeding program for transferring elite traits from wild relatives (Cortés et al., 2020)”.

Revised portions are shown in line 309-315.

References:

Cortés, A.J.; López-Hernández, F. Harnessing crop wild diversity for climate change adaptation. Genes 2021, 12, 783.

Cortés, A.J.; Restrepo-Montoya, M.; Bedoya-Canas, L.E. Modern strategies to assess and breed forest tree adaptation to changing climate. Front. Plant Sci. 2020, 11, 583323.

Point 5: In this regard, authors should expand the discussion on what is the potential contribution for soybean breeding of last-generation approaches such as genomic prediction (i.e. Front Plant Sci 2021 12:624156 in soybean, reviewed more widely here Front Plant Sci 2020 11:583323, citation to be included), genome-wide scans of selection signatures (carefully review by Front Genet 2020 11:564515), machine learning (nicely discussed in Trends Genet 2018 34:301-12, review to be included), and speed breeding (recently covered in Trend Genet 2021 37:1124-36, review also to be included).

Response 5: Thanks for the comments. Those above strategies such as genomic prediction and machine learning could accelerate the breeding efficiency of complex traits, like yield and plant architecture (Cortés et al., 2020; Schrider et al., 2018; Shi et al. 2021). The effort of them rely on many factors, such as perfect genomic information, accurate phenotypic (or phenomics) data, appropriate algorithms and representative germplasm. Hence, enhanced interoperability between different omics and phenotyping platforms, leveraged by evolving machine learning tools, will help provide explanations for complex plant traits (Varshney et al., 2021). For soybean VG and PH traits, QTLs with relatively larger effect are involved, indicating they are candidates that can be manipulated by MAS or genome editing (Trend Genet 2021 37:1124-36).

References:

Cortés, A.J.; López-Hernández, F.; Osorio-Rodriguez, D. Predicting thermal adaptation by looking into populations' genomic past. Front. Genet. 2020, 11, 564515.

Schrider, D.R.; Kern, A.D. Supervised machine learning for population genetics: a new paradigm. Trends Genet 2018 34:301-312

Shi, A.N.; Gepts, P.; Song, Q.J.; Xiong, H.Z.; Michaels, T.E.; Chen, S.Y. Genome-wide association study and genomic prediction for soybean cyst nematode resistance in USDA common bean (Phaseolus vulgaris) core collection. Front. Plant Sci. 2021, 12, 624156.

Varshney, R.K.; Bohra, A.; Roorkiwal, M.; Barmukh, R.; Cowling, W.A.; Chitikineni, A.; Lam, H.M.; Hickey, L.T.; Croser, J.S.; Bayer, P.E.; et al. Fast-forward breeding for a food-secure world. Trends Genet. 2021, 37, 1124-1136.

Point 6: The feasibility of these approaches must explicitly be revisited in a new perspectives section, just after the current discussion section. This new section should start by discussing potential caveats of this and previous studies until now, and propose new avenues of research. Authors should envision key experimental, and novel analyses that will assist in the near future further studies on soybean genetic mapping and improvement of plant architecture traits. These sections will facilitate readers to highlight major trends and opportunities.

Response 6: Thanks for the suggestion. The GWAS, MAS and genome editing techniques were discussed in a new perspective section for plant architecture traits improvement. New content was supplemented in penultimate paragraph “To better elucidate the genetic relationships throughout soybean domestication, efforts are needed to identify more loci related with plant architecture to enhance the utilization of wild soybean resources. Genome-wide association study (GWAS) should be used for VG analysis based on evaluation of phenotype and genotype of representative soybean panel as that was conducted for soybean cyst nematode resistance (Shi et al., 2021). To clone the genes underlying these loci, chromosome segment substitution lines (CSSL) population should be constructed for fine mapping of each related QTL (Li et al., 2019). Followed precise genome editing techniques will benefit the utilization of genes control plant architecture in selection of ideal varieties (Varshney et al., 2021), provide strategies to reduce gene pleiotropy through modifying cis-regulatory regions (Song et al., 2022), or manipulate target genes for de novo domestication of wild crop relative (Yu et al., 2021)”.

Revised portions are shown in line 316-326.

References:

Shi, A.N.; Gepts, P.; Song, Q.J.; Xiong, H.Z.; Michaels, T.E.; Chen, S.Y. Genome-wide association study and genomic prediction for soybean cyst nematode resistance in USDA common bean (Phaseolus vulgaris) core collection. Front. Plant Sci. 2021, 12, 624156.

Li, R.C.; Jiang, H.W.; Zhang, Z.G.; Zhao, Y.Y.; Xie, J.G.; Wang, Q.; Zheng, H.Y.; Hou, L.L.; Xiong, X.; Xin, D.W.; et al. Combined linkage mapping and BSA to identify QTL and candidate genes for plant height and the number of nodes on the main stem in soybean. Int. J. Mol. Sci. 2019, 21, 42.

Varshney, R.K.; Bohra, A.; Roorkiwal, M.; Barmukh, R.; Cowling, W.A.; Chitikineni, A.; Lam, H.M.; Hickey, L.T.; Croser, J.S.; Bayer, P.E.; et al. Fast-forward breeding for a food-secure world. Trends Genet. 2021, 37, 1124-1136.

Song, X.G.; Meng, X.B.; Guo, H.Y.; Cheng, Q.; Jing, Y.H.; Chen, M.J.; Liu, G.F.; Wang, B.; Wang, Y.H.; Li, J.Y.; et al. Targeting a gene regulatory element enhances rice grain yield by decoupling panicle number and size. Nat. Biotechnol. 2022, online ahead of print. https://doi.org/10.1038/s41587-022-01281-7.

Yu, H.; Lin, T.; Meng, X.B.; Du, H.L.; Zhang, J.K.; Liu, G.F.; Chen, M.J.; Jing, Y.H.; Kou, L.Q.; Li, X.X.; et al. A route de novo domestication of wild allotetraploid rice. Cell 2021, 184, 1156-1170.

Round 2

Reviewer 2 Report

Thanks for the careful amendments, with which I agree. The work is suitable to proceed its editorial workflow.